# A New Efficient Method For Combining Gradients Of Different Orders
# Conference Submissions

## Abstract

We present a new optimization method called GOC(Gradient Order Combination) which a combination based on the products of Hessian matrices of different orders and the gradient. the parameter r (the recipprocal of steplenth) is taken as analysis target, we can regard the SD method as a first-order and the CBB method as second-order. Whave developed third-order and even higher-order, which offer faster convergence rates.

## 1 Introduction

In this paper, we consider the the unconstrained optimization problem with convex quadratic form

$$minf(x) = \frac{1}{2}x^T A x - b^T x \tag{1}$$

where $x \in \mathbb{R}^n$, $b \in \mathbb{R}^n$, $A \in \mathbb{R}^{n \times n}$ is a symmetric and positive definite matrix.

The common solution methods for solving Eq(1) are iterarive methods of the following form

$$x_{k+1} = x_k - \alpha_k \nabla f(x_k) \tag{2}$$

$$f(x_{k+1}) = f(x_k - \alpha_k \nabla f(x_k)) \tag{3}$$

where $\alpha_k$ is a steplenth,gradient descent method and its variants are the most common optimization method.for GD method,if we minimizes Eq.(3) with exact line search,then we get

$$\alpha_k^{SD} = \frac{\nabla f_k^T \nabla f_k}{\nabla f_k^T A \nabla f_k} = \frac{g_k^T g_k}{g_k^T A g_k} \tag{4}$$

this method proposed by A.Cauchy (1847) is called steepest descent method,so $\alpha_k^{SD}$ is also called Cauchy step length. the method's convergence rate is very sensitive to ill condition number and may be very slow ,when the f(x) is quadratic $x_k$ will satisfy the

$$\frac{f(x_{k+1}) - f(x^*)}{f(x_k) - f(x^*)} \leq (\frac{\lambda_1 - \lambda_n}{\lambda_1 + \lambda_n})^2 \tag{5}$$

During the iteration process, the SD method exhibits a zigzag phenomena which was explained by Akaike (1959) , J.BARZILAI & J.M.BORWEIN (1988) proposed a nonmonotone steplength which certain quisi-Newton method, it has two choice for $a_k$,respectively:

$$\alpha_k^{BB1} = \frac{s_{k-1}^T s_{k-1}}{s_{k-1}^T y_{k-1}} \tag{6}$$

$$\alpha_k^{BB2} = \frac{s_{k-1}^T y_{k-1}}{y_{k-1}^T y_{k-1}} \tag{7}$$

where $s_{k-1} = x_k - x_{k-1}$ and $y_{k-1} = g_k - g_{k-1}$,the BB step can be seen Cauchy step with previous iteration. Barzilai and Borwein proved R-superlinear convergence rate in two dimension.Yuan (2008) for general n dimensional convex quadratic case, the method is convergent too and has a properties of R-linear rate of convergence. there are some optimization methods based on gradient, YH (2003) decrease the gradient norm , Yuan (2006) and YH (2005) design a alternate steps , in two dimension case, it could convergence 3 steps. Serafino;F.Riccio;G.Toraldo (2013) propose SDA with a fixed stepsiz in sussesive steps. and SDC (R.De Asmundisdi SerafinoD (2014)) adding Cauchy step comparing SDA. Sun C (2020) propose new step size based on Cauchy stepsize. Z (2015) select random stepsize at some range.Raydan M (2002) introduce RSD which accelerates convergence by introducing a relaxation parameter between 0 and 2 in the standard Cauchy method, they also propose CBB method which is a combination of the SD and BB method ,the CBB algorithm is much more efficient than BB method.In this paper, we construct a new descent method by combining the gradient with products of the Hessian matrix of different orders.

## 2 ANALYSIS OF SD AND CBB METHODS

From Eq(4), we define a parameter $r_k$ as follows:

$$r_k = \frac{1}{2\alpha_k} = \frac{g_k^T A g_k}{2 g_k^T g_k} \tag{8}$$

the initial point is $x_0$ we set

$$x_0^s = x_0 - \alpha_0 g_0 \tag{9}$$

It is evident that $x_0^s$ is the result obtained after applying the steepest descent method.than we search in the $A g_0$ direction and find the point $x_1^A$, the vecotrs $\overrightarrow{x_0 x_0^s}$ and $\overrightarrow{x_0 x_1^A}$ are perpendicular. than we discover the symmetric points $x_1$ in the direciton $\overrightarrow{x_1^A x_0}$,it is obvious $|x_1 x_0^s| = |x_0^s x_0^A|$,as shown in Fig(1). In order to make the analysis more convenient and intuitive,considering a situation the objective function is a simple n dimensions hyper-ellipsoid stimulating Eq(1)

$$f(x) = \sum_{i=1}^n a^{(i)} x^{(i)^2} \tag{10}$$

$$r = \frac{\sum_{i=1}^n a^{(i)^3} x^{(i)^2}}{\sum_{i=1}^n a^{(i)^2} x^{(i)^2}} = \frac{\sum_{i=1}^n a^{(i)} g^{(i)^2}}{\sum_{i=1}^n g^{(i)^2}} \tag{11}$$

where $0 < a^{(n)} \le a^{(n-1)} \le ...... \le a^{(1)}$,$g^{(i)} = 2a^{(i)} x^{(i)}$, the initial point $x_0 = [x_0^{(1)}, x_0^{(2)}, ......x_0^{(n)}]$

from Eqs.(9) and (10),we have

$$x_0^s = x_0 - \frac{\nabla f(x_0)}{2r_0} \tag{12}$$

$$x_0^{s(i)} = x_0^{(i)}(1 - \frac{a^{(i)}}{r_0}) \tag{13}$$

we define $v_0 = A g_0$ , $l_0 = \|x_0 x_0^s\|$ , $l_0^A = \|x_0 x_0^A\|$, $\theta_0$ is the angle between $g_0$ and $v_0$ we have

$$cos\theta_0 = \frac{g_0^T v_0}{\|g_0\| \|v_0\|} = r_0 [\frac{\sum_{i=1}^n a^{(i)^2} x_0^{(i)^2}}{\sum_{i=1}^n a^{(i)^4} x_0^{(i)^2}}]^{\frac{1}{2}} \tag{14}$$

$$\frac{\|v_0\|}{\|g_0\|} = 2 [\frac{\sum_{i=1}^n a^{(i)^2} x_0^{(i)^2}}{\sum_{i=1}^n a^{(i)^4} x_0^{(i)^2}}]^{\frac{1}{2}} \tag{15}$$

$$l_0^A = l_0/cos\theta_0 = \frac{(\sum_{i=1}^n a^{(i)^4} x_0^{(i)^2})^{\frac{1}{2}}}{r_0^2} \tag{16}$$

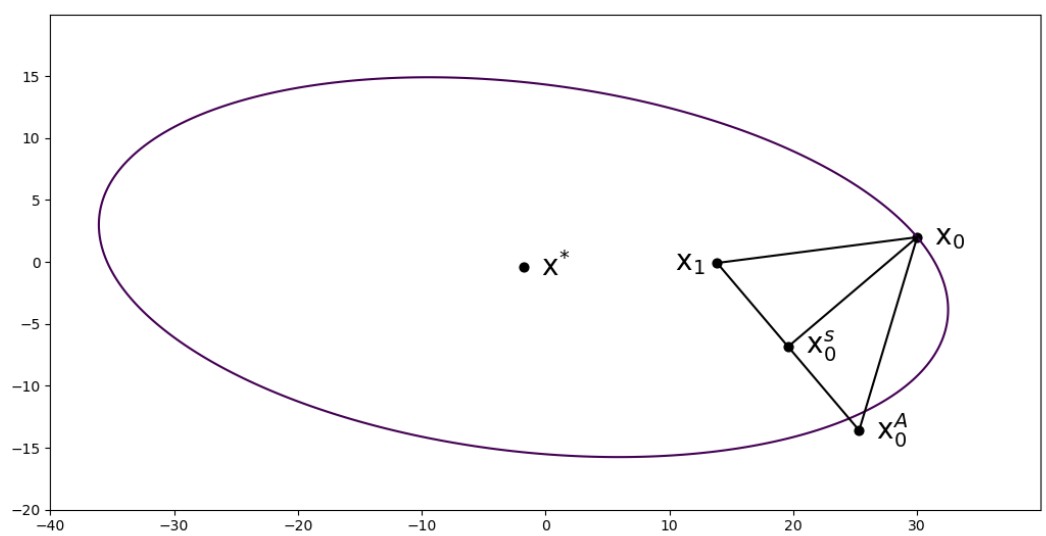

Figure 1: SD and CBB

we have

$$x_0^{A^{(i)}} = x_0^{(i)}[1 - (\frac{a^{(i)}}{r_0})^2]$$ (17)

because $x_1$ is a symmetric points of $x_0^A$ about $x_0^s$, so $x_1 = 2x_0^s - x_0^A$

$$x_1^{(i)} = x_0^{(i)}(1 - \frac{a^{(i)}}{r_0})^2 = x_0^{(i)}\mu_0^{(i)^2}$$ (18)

It is evident that $x_1$ is the result obtained after applying the CBB method which is equivalent to using SD method with the same steplenth in two consecutive iterations. From the above analysis and Figure(1), we can see that the CBB update direction is symmetric to the $Ag$ direction with the current gradient as the axis.

## 3 GOC METHOD

We consider a sequence of m consecutive identical step sizes as the update point,assuming the current point is $x_k$,we can obtain the values of $r$ for the three points $x_k$,$x_k^s$, and $x_{k+1}$ as follows:

$$r_k = \frac{\sum_{i=1}^{n} a^{(i)} g_k^{(i)^2}}{\sum_{i=1}^{n} g_k^{(i)^2}}$$ (19)

$$r_k^s = \frac{\sum_{i=1}^{n} a^{(i)} g_k^{(i)^2} \mu_k^{(i)^2}}{\sum_{i=1}^{n} g_k^{(i)^2} \mu_k^{(i)^2}} = \frac{\sum_{i=1}^{n} a^{(i)} g_k^{(i)^2} [r_k - a^{(i)}]^2}{\sum_{i=1}^{n} g_k^{(i)^2} [r_k - a^{(i)}]^2}$$ (20)

$$r_{k+1} = \frac{\sum_{i=1}^{n} a^{(i)} g_k^{(i)^2} \mu_k^{(i)^{4m}}}{\sum_{i=1}^{n} g_k^{(i)^2} \mu_k^{(i)^{4m}}} = \frac{\sum_{i=1}^{n} a^{(i)} g_k^{(i)^2} [r_k - a^{(i)}]^{4m}}{\sum_{i=1}^{n} g_k^{(i)^2} [r_k - a^{(i)}]^{4m}}$$ (21)

we will analyses several situations for diffrent initial values and the effect of r.

1.if the current point $x_k$ lies in the larger eigenvalue direction, the gradient value of the lager eigenvector is biger also, the larger eigenvalue component account for a greater proportion,so $r_k$ tend to $a^{(1)}$,the value of $a^{(i)}$ direction near the $r_k$ will fall sharply ,$r_{k+m}$ and $r_k^s$ will move to $a^{(n)}$ direction .in this case,$\mu^{(i)4m} < \mu^{(i)^2}$,$r_{k+1} < r_k^s$.from Figure(2a),the bigger m value is and the relatively smaller the $\mu$ value and the faster the decrease rate of different eigenvalue direction.

2.if the current point $x_k$ have a huge value at the minimize eigenvector direction compared to other eigenvector directions,$r_k$ tend to $a^{(n)}$,the great majority of $a^{(i)}$ direction value are much larger than r value. so $\mu^{(i)}$ is far larger than 1 especially in the direction of large eigenvale. from Figure(2b), the bigger m value is and the relatively bigger the $\mu$ value , there has a sharp rise in the bigger eigenvalue direction.

3. considering more general cases, the distribution of the current point $x_k$ component are random, the $r_k$ is random value between $a^{(1)}$ and $a^{(n)}$ correspondingly.those eigenvetor direction value agree with $r_k$ will decrease more quickly. $r_{k+1}$ and $r_k^s$ will become larger and smaller according to $r_k$.if the $r_k$ value is in the middle area of eigenvalues. from Figure(2c), the larger $\mu$ value signify faster decreas rate like Figure(2a).

Based on the analysis above, r value will seesaw between larger eigenvalue area and smaller eigenvalue area generally.the component of small eigenvalue determine the convergence rate and is hard to reduce .for SD method,r will stabilize in two certain value which means to be relatively fixed decrase rate. Comparing the SD method ,the CBB method's r value have more wider range change , and have higher descent rate in the direction of small eigenvalue also. Randan and Svaiter have proven that the sequence $x_k$ generate by CBB method converages Q-linerarly in the norm with convergence factor $1 - \theta = \frac{\lambda_{max} - \lambda_{min}}{\lambda_{max}}$

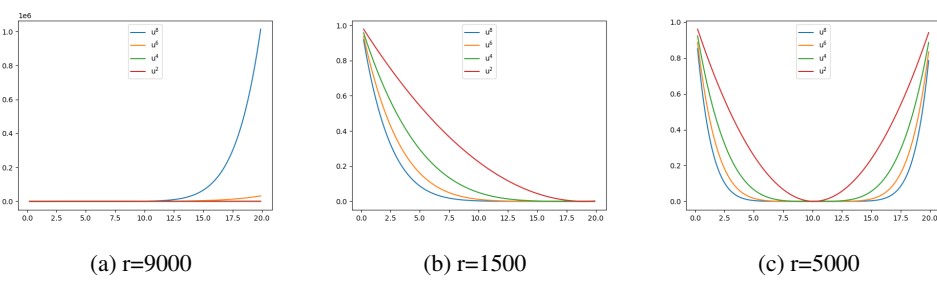

(a) r=9000          (b) r=1500          (c) r=5000

Figure 2: $x \in [0.1, 10000]$, $\mu = 1 - \frac{x}{r}$

It can be seen that if we analyze from the perspective of eigenvalue, we will find that the SD method is first-order, while the CBB method is second-order.and we can develop methods of higher order.

Assuming it is of order m, we have

$$x_1^{(i)} = x_0^{(i)}(1 - \frac{a^{(i)}}{r_0})^m = x_0^{(i)} \sum_{i=1}^{n} C_m^k(-\mu_0^{(i)})^k \tag{22}$$

where $\mu_0^{(i)} = \frac{a^{(i)}}{r_0}$

We konw that applying the Hessian-free method to a vector $v$ is equivalent to multiply $v$ by $A$ ,If the vector is the gradient, then each application of the Hessian-free method is equivalent to multiplying each component of the eigenvector by its corresponding eigenvalue. so $\mu_0^{(i)^k}$ is equivalent to applying the Hessian-free method $k$ times.so by combining different numbers of Hessian-free method iterations, we can achieve Eq(22). if we take m to be 3. then we have

$$x_1^{(i)} = x_0^{(i)}(1 - \frac{a^{(i)}}{r_0})^3 = x_0^{(i)}(1 - 3\mu_0^{(i)} + 3\mu_0^{(i)^2} - \mu_0^{(i)^3}) \tag{23}$$

We transform the above equation into another form as follow

$$x_1 = x_0 - 3\frac{g_0}{r_0} + 3\frac{Ag_0}{r_0^2} - \frac{A^2 g_0}{r_0^3} \tag{24}$$

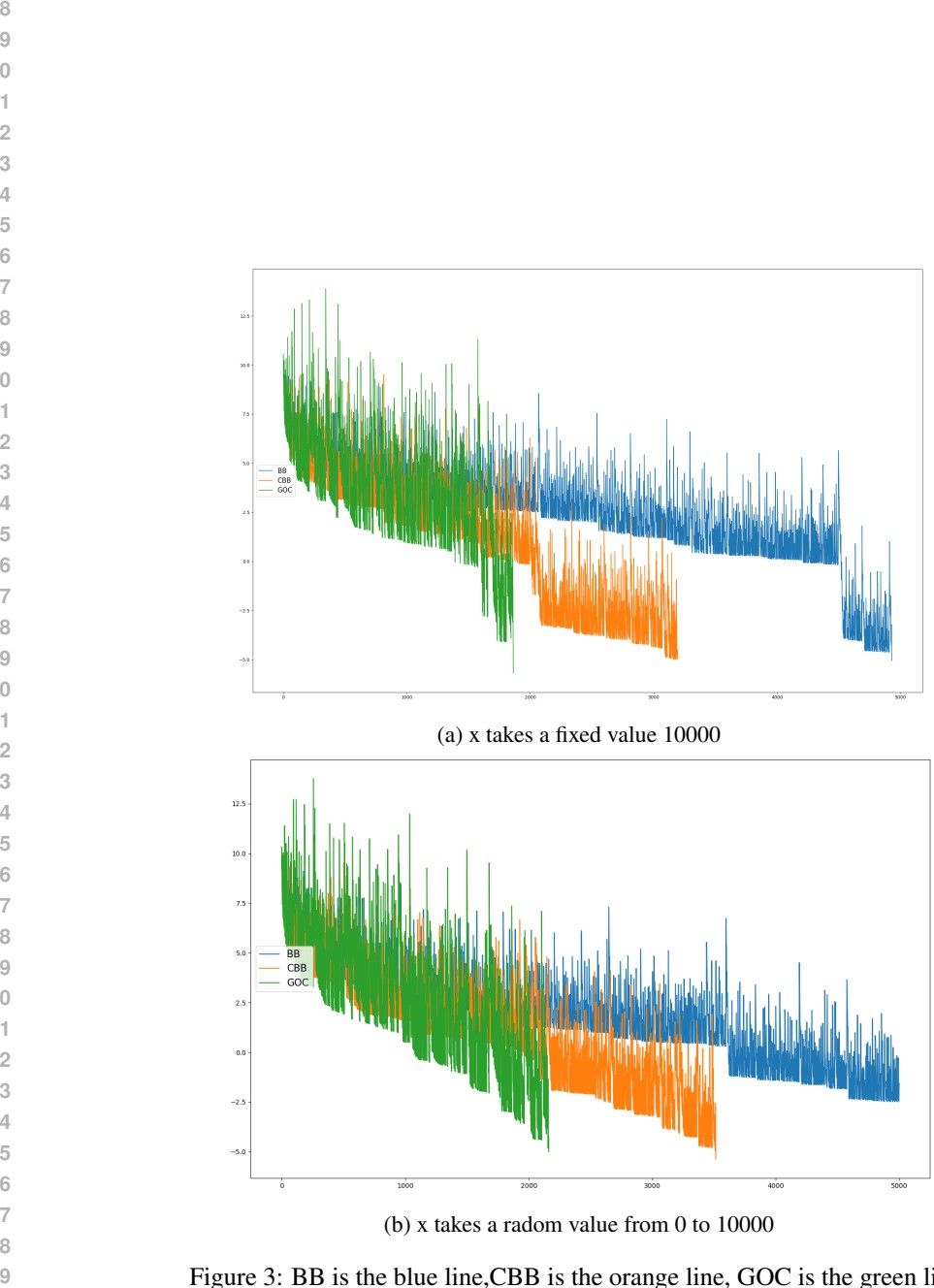

(a) x takes a fixed value 10000

(b) x takes a radom value from 0 to 10000

Figure 3: BB is the blue line,CBB is the orange line, GOC is the green line.

From the above equation, it can be seen that if we calculate $Ag_0$ and $A^2g_0$, we can obtain the final updated value. We first compute the gradient at the current point $g_0$,and then move along the current negative gradient with a step size of $d$ to reach the point $x_0^1$.At $x_0^1$,we compute gradient $g_0^1$ and the value of $r_0$ .By calculating $g_0-g_0^1$,we obtain $dAg_0$. Next, we move along the direction of the gradient $g_0^1$ with a step size of $d$ to reach the point $x_0^2$,and compute the gradient $g_0^2$,By calculating $g_0 - g_0^2$,we obtain $d^2A^2g_0$.In this way, we obtain all the values in Eq(24), thereby obtaining the updated value $x_1$. From the above, it can be seen that by updating once in the negative gradient direction and once in the positive gradient direction with a fixed step size $d$, we can calculate the final updated point.

---

**Algorithm 1** Gradient Order Combination Algorithm

---

**Require:** $f(x)$: objective funtion; $x_0$: initial solution; $d$: step size; $\varepsilon$: objective gradient norm value
**Ensure:** optimal $x^*$
  initial $x_0$
  **while** $(|f(x_k|) > \varepsilon)$ **do**
    compute $x_k$ gradient $g_k$;
    compute $x_k^1 = x_k - dg_k$;
    compute $x_k^1$ gradient $g_k^1$ and $r_k$;
    compute $Ag_k = \frac{g_k-g_k^1}{d}$;
    compute $x_k^2 = x_k^1 + dg_k^1$;
    compute $x_k^2$ gradient $g_k^2$;
    compute $A^2g_k = \frac{g_k-g_k^2}{d^2}$;
    compute $x_{k+1} = x_k - \frac{3g_k}{r_k} + \frac{3A^2g_k}{r_k^2} - \frac{A^3g_k}{r_k^3}$
  **end while**

---

## 4 NUMERICAL EXPERIMENTS

Considering an example as follow

$$f(x) = \sum_{i=1}^{100000} a^{(i)}x^{(i)^2} \tag{25}$$

the sequence $a^{(i)}$ is arithmetic progression and $0.001 \leq a^{(i)} \leq 10000$,$x_0^{(i)}$ is a fixed value of 10000.the stopping parameter $\epsilon = 10^{-5}$.we perform 5000 iterations caculation by menas of three method(BB CBB GOC).for demonstration on the figure, the norm value is processed with logarithm in order to limit big changing range of data. The number of times that BB satisfies the stopping condition is 4930,the CBB method is 3194,The GOC method is 1864.as shown in Figure(3a) we set $x_0^{(i)} = 10000 * rd$, $rd$ is randomly generated in (0,1),The maximum value of $x_0^{(i)}$ is 9999.9531,and the minimum value of $x_0^{(i)}$is 0.0423.The number of times that CBB method satisfies the stopping condition is 3515,The GOC method is 2163,and the BB method could not satisfy the stop condition.as shown in Figure(3b)

## 5 CONCLUSION

We conducted an in-depth analysis of the SD method and the CBB method from the perspective of optimal step size. We found that they can be regarded as methods of different orders within the same pattern. Based on this, we designed a higher-order method, which demonstrates a faster rate of descent.

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
