# OpenReview forum: "A New Efficient Method For Combining Gradients Of Different Orders"
_ICLR.cc/2026/Conference — Submitted to ICLR 2026_

### Official Review · Reviewer_UWoS · 2025-10-16

**Soundness:** 1
**Presentation:** 1
**Contribution:** 1
**Rating:** 0
**Confidence:** 4

**Summary:**

This paper proposes an optimization algorithm called the Gradient Order Combination (GOC) method. The key idea is to regard the Steepest Descent (SD) and Cauchy–Barzilai–Borwein (CBB) methods as lower-order cases of a more general gradient-based framework, and to extend this framework to higher orders in order to accelerate convergence. The authors analyze quadratic optimization problems in the eigenvalue space and show that increasing the order of the method reduces sensitivity to the condition number and alleviates variation in convergence rates across different directions. Numerical experiments demonstrate that the proposed method converges with significantly fewer iterations than the SD, BB, and CBB methods. Overall, the paper provides a unified perspective on existing gradient methods and presents an effective generalization that achieves faster convergence without additional computational cost.

**Strengths:**

The paper attempts to extend the classical Steepest Descent, Barzilai–Borwein (BB), and Cauchy–Barzilai–Borwein (CBB) methods within a unified framework, and proposes the GOC method, which aims to automatically adjust convergence behavior across different directions according to the eigenvalue spectrum of the Hessian, thereby achieving faster convergence.

**Weaknesses:**

The optimization problem analyzed in this paper is overly simple, focusing only on an unconstrained convex quadratic function that corresponds to a textbook-level example rather than a genuine research problem. Because of this limited setting, the proposed method provides little theoretical or practical insight.

The claimed novelty of combining gradients of different orders appears largely superficial. The proposed GOC algorithm is mathematically very close to existing Barzilai–Borwein and Conjugate Gradient methods, essentially representing a minor reformulation rather than a new concept. No new theoretical framework or general convergence analysis is provided to distinguish it from these established approaches.

Theoretical analysis is confined to the simple quadratic case (as in Eq. 10) without any discussion or experiments for more general convex or nonconvex problems. Comparisons are limited to basic SD and BB-type methods, which is not sufficient to establish the research contribution of the paper.

The paper repeatedly claims that the proposed method is efficient, but it does not analyze or quantify the computational cost per iteration. A formal comparison with the cost of SD or CBB methods would be needed to support this claim.

Furthermore, the presentation lacks clarity: many symbols are introduced without explicit definition, making it difficult to follow the derivations. Figures and captions are poorly explained and do not clearly support the claims. Finally, both the English writing and the logical organization fall short of the standard expected for a research paper.

**Questions:**

1. **Scope and Generality of the Problem Setting**:
   The analysis in this paper is limited to an unconstrained convex quadratic problem. Could the authors explain whether the proposed method can be applied to more general convex or nonconvex functions? If so, what theoretical properties such as convergence guarantees or stability would still hold? A clear description of the intended application domain would make the contribution easier to understand.

2. **Relation to Existing Methods**:
   The proposed GOC method appears very similar to classical Steepest Descent, Barzilai Borwein, and Conjugate Gradient methods. The paper does not provide any explanation of the computational cost per iteration. It would be helpful to clarify how the cost compares with that of the SD and CBB methods, in order to support the claim of efficiency.

3. **Novelty and Theoretical Contribution**:
   The paper claims that increasing the order leads to faster convergence. Can this claim be justified theoretically for functions beyond the quadratic case? For example, can the authors show any formal convergence rate or bound that demonstrates an advantage over existing methods?

4. **Numerical Experiments and Evaluation**:
   The experiments focus only on very simple quadratic problems. To evaluate the usefulness of GOC, it would be important to include tests on more general optimization problems, such as logistic regression or smooth nonconvex objectives. Why were the comparisons restricted to SD and BB type methods? Including modern accelerated or adaptive methods would make the results more convincing.

5. **Clarity of Presentation**:
   Many mathematical symbols are introduced without clear definition, which makes the derivations difficult to follow. Please ensure that every symbol is defined when it first appears and that the notation is consistent throughout. The captions of the figures are generally insufficient. Each caption should clearly describe what the figure represents and explain how it relates to the theoretical discussion.

6. **Writing and Organization**:
   The English writing and the logical flow are difficult to follow in several parts. Careful editing and reorganization of the paper would greatly improve readability. In addition, the manuscript requires thorough proofreading for grammar, usage, articles, and tense consistency. The introduction should more clearly state the motivation, the gap in existing research, and the specific contribution of the proposed method.

---

### Official Review · Reviewer_hvsU · 2025-10-23

**Soundness:** 1
**Presentation:** 1
**Contribution:** 1
**Rating:** 0
**Confidence:** 5

**Summary:**

The proposer GOC method is as to be a “higher-order” gradient method with “faster convergence rates.”  However, this is just a heuristic of fixed-step GD with
- No theorems or proofs support any rate claims
- The method is just $m$-repeated gradient-descent steps with the same step size written as a polynomial update.

**Strengths:**

not applicable

**Weaknesses:**

# The method
The proposed GOC(3) update (Eq. 24, p. 4):
$x_{k+1}=x_k-\frac{3g_k}{r_k}+\frac{3Ag_k}{r_k^2}-\frac{A^2 g_k}{r_k^3}$
is *exactly* what you get if you run **three consecutive GD steps** with a fixed step size $1/r_k$ on a quadratic function
There is no new principl -- it’s the degree-3 expansion of $(I-\tfrac{A}{r})^3$.

For SPD $A$ $x_{k+1}-x^* = (I - \tfrac{A}{r})^m (x_k - x^*)$ i.e. the same as **m fixed-step GD updates**.
Each “higher-order” update costs ~m gradient evaluations (to build $A^j g$), yielding no gain per gradient call.

A true rate statement (not in paper): $\|e_{k+m}\|_A \le \rho^m \|e_k\|_A,\quad
\rho = \max_i |1 - \alpha \lambda_i|$ Nothing faster than plain GD; no acceleration mechanism like Chebyshev or Nesterov is shown.


The “Hessian-free” part (Algorithm 1, p. 6) estimates $A g_k, A^2 g_k$ via finite differences of gradients, but that simply re-computes what three GD steps would do at higher cost.

# Claims are not supported
- Faster convergence rate” --  no theorem, lemma, or proof
- “Third-order / higher-order”  -- not applicable, the method uses repeated GD with same step size
- “Combines gradients and Hessian powers” --  only valid for quadratics; uses finite differences otherwise
- “CBB is second-order” --  incorrect, equates to “two SD steps with same step”
- “Outperforms BB/CBB” -- only wo toy plots (Fig. 3 p. 5), unlabeled axes, (remember the method is just 3 GD step, so it can be three times faster in iterations, no surprize)

# Technical correctness problems
- Definition inconsistency --  Eq. (8) defines $r_k = \frac{1}{2\alpha_k}$, but later updates use $x^+ = x - \frac{1}{r}g$. The factor-of-2 mismatch propagates.
- Eq. (22) -- incorrect summation index; mixes dimension $n$ and polynomial degree $m$
- Algorithm 1 computes $ x_{k+1}=x_k-\frac{3g_k}{r_k}+\frac{3A^2 g_k}{r_k^2}-\frac{A^3 g_k}{r_k^3}$,  which disagrees with Eq. (24): wrong powers and coefficients
- Hessian-free equals multiplication by A statement is only true for constant Hessians (quadratics). outside that case, gradient-difference estimates are biased, no error analysis given.
- Notation chaos $λ_1, λ_n, a^{(i)}$ used interchangeably; undefined symbols; factors of 2 appear/disappear between sections.

# Experiments
- Two synthetic cases only
- Axes unlabeled; “norm value processed with logarithm” without specifying
- “Number of times stopping condition satisfied is 4930” -- meaningless
- No runtime or gradient-count comparison
- No baselines like Conjugate Gradient, which dominates for quadratics.
- Not reproducible.

# Writing and formatting quality
- Misspellings throughout (“recipprocal”, “steplenth”, “menas”, etc.).
- Unclear grammar and missing articles.
- Equations (1)–(25) mix inconsistent notation for the same quantities.
- Figures lack labels; Fig. 1 (p. 3) is unlabeled geometry sketch; Fig. 3 plots have no axis units.
- Algorithm 1 pseudo-code uses inconsistent variable names (d2, g1k, g2k).

**Questions:**

yes, for AC can i somehow complaint about the quality of the the paper?

---

### Official Review · Reviewer_ngqp · 2025-10-29

**Soundness:** 2
**Presentation:** 1
**Contribution:** 2
**Rating:** 2
**Confidence:** 4

**Summary:**

This paper introduces a new optimization method called GOC, which interprets the reciprocal of the step size $r = 1/(2\alpha)$ as a unifying parameter for Steepest Descent (SD) and cyclic Barzilai–Borwein (CBB).

By extending this view, the authors propose an “m-th order” generalization and implement the case m=3 using finite differences to approximate Hessian–gradient products. Experiments on diagonal quadratic problems show that GOC requires fewer iterations than BB and CBB.

**Strengths:**

The idea of interpreting repeated updates as a polynomial acceleration is theoretically interesting. The method is Hessian-free and seems simple to implement.

**Weaknesses:**

1, Inconsistency between Eq. (24) and Algorithm 1: the algorithm uses $A^3 g_k$ (Line 291) which is never computed, contradicting the derived formula (24).

2, Claims of higher efficiency are misleading since each iteration requires ~3 gradient evaluations; total computational cost is not compared.

3, Theoretical analysis and convergence guarantees are missing. Actually, this paper lacks a rigorous convergence analysis. Although it cites the Q-linear convergence of the CBB method, it provides no convergence rate, complexity bounds, or even a proof of convergence for GOC. The analysis in Section 3 is heuristic.

4, Experiments are limited to diagonal quadratic problems with no tests on general or non-quadratic objectives.

5, The paper contains numerous typos even in the abstract (e.g., “recipprocal,” “steplenth,” “Whave”).

6, The notion of “m-th order” is not mathematically well-defined. The paper informally associates SD with first order, CBB with second order, and GOC with third order, but does not formally establish what “order” means in this context. Usually, the order means the degree of derivate information used, but in this paper, this interpretation does not apply.

**Questions:**

1. Which version of the update rule (Eq. 24 or Algorithm 1) was actually used in experiments?
2. How is $A^3 g_k$ computed or approximated?
3. Have you compared total gradient evaluations rather than just iteration counts?
4. Does the GOC framework extend beyond diagonal quadratic cases?

---

### Official Review · Reviewer_ccXH · 2025-10-31

**Soundness:** 1
**Presentation:** 1
**Contribution:** 1
**Rating:** 0
**Confidence:** 5

**Summary:**

This paper proposes a new optimization algorithm, **Gradient Order Combination (GOC)**, which aims to generalize gradient-based methods such as Steepest Descent (SD) and the Cauchy–Barzilai–Borwein (CBB) method by combining gradients and Hessian–vector products of different orders. The authors derive a “third-order” update rule and show numerical experiments on simple synthetic quadratic problems, claiming faster convergence than SD, BB, and CBB.

**Strengths:**

- The paper’s **overall motivation** to create a unified perspective connecting first- and second-order gradient methods, is conceptually interesting.
- The derivation tries to provide **geometric intuition** linking SD and CBB methods.

**Weaknesses:**

- **Extremely poor writing and structure:** The paper is very difficult to read due to pervasive grammatical errors, unclear explanations, and inconsistent notation. It does not meet the presentation quality expected from this venue.
- **Lack of novelty:** The proposed method is a heuristic combination of known concepts (Barzilai–Borwein steps and Hessian-free approximations). The claimed “order” generalization is not theoretically justified or novel in the optimization literature.
- **No theoretical analysis:** There are no convergence proofs, rate guarantees, or complexity discussions. The analysis is entirely heuristic.
- **Internal inconsistencies:**
  - Equation (24) and Algorithm 1 do **not match** the algorithm uses higher powers of \(A\) than the derivation specifies.
  - The definition and interpretation of \(r_k\) (as reciprocal step length) are inconsistent across the text.
  - The stopping criterion in the algorithm contradicts the description in the text.
- **Weak experiments:** Only trivial convex quadratic examples are tested, with vague reporting (“number of times satisfying the stopping condition”). No comparisons are made to standard baselines such as Conjugate Gradient, L-BFGS, or Nesterov acceleration.
- **No evidence of practicality:** The algorithm requires multiple gradient evaluations per iteration, increasing cost without demonstrating real advantages.

**Questions:**

Why waste our time?

---

### Meta-Review · Area_Chair_kzW8 · 2026-01-04

**Summary:**

This paper proposes an optimization algorithm, which aims to generalize gradient-based methods such as Steepest Descent and the Cauchy–Barzilai–Borwein method by combining gradients and Hessian–vector products of different orders.

**Reviewer Concerns:**

1. Theoretical/Novelty Gaps: All reviewers noted no convergence proofs, rate guarantees, or complexity bounds. GOC equals m fixed-step GD steps (hvsU) and is a minor reformulation of BB/CG (UWoS); "m-th order" lacks definition (ngqp).
2. Technical Inconsistencies: Eq. (24) and Algorithm 1 mismatch (e.g., \(A^3 g_k\) in code vs. \(A^2 g_k\) in derivation); \(r_k\) has a factor-of-2 error (hvsU); notation is chaotic.
3. Weak Experiments: Only trivial quadratic problems tested; no standard baselines (CG, L-BFGS); unlabeled figures, no cost analysis (ngqp, hvsU).
4. Presentation: All rated "poor" (1/5) due to typos, undefined symbols, and uncaptioned figures.
5. Unanswered Question: Key questions (e.g., GOC’s non-quadratic applicability, \(A^3 g_k\) computation) remain unaddressed.

**Reviewer Scores:**

Remain the raw score.

---

### Decision · Program_Chairs · 2026-01-26

Reject